# Visual Impairment in Hemodialyzed Patients—An IVIS Study

**DOI:** 10.3390/medicina59061106

**Published:** 2023-06-07

**Authors:** Leszek Sułkowski, Mateusz Rubinkiewicz, Andrzej Matyja, Maciej Matyja

**Affiliations:** 1Department of General Surgery, Regional Specialist Hospital, 42-218 Częstochowa, Poland; leszeksulkowski@szpitalparkitka.com.pl; 22nd Department of General Surgery, Jagiellonian University Medical College, 30-688 Kraków, Poland

**Keywords:** Impact of Visual Impairment Scale, chronic kidney disease (CKD), quality of life, life satisfaction, World Health Organization Quality of Life questionnaire (WHOQOL-BREF), Cantril Ladder

## Abstract

*Background and Objectives*: The growing and aging population of hemodialysis patients has become increasingly disabled, with more complex comorbidities, and are older upon initiating dialysis. Visual impairment can adversely affect their quality of life and life satisfaction. Treatment evaluation should not only consider remission of the disease, but also the improvement of quality of life and life satisfaction. This is a single-center cross-sectional study. It was designed to evaluate visual impairment in hemodialyzed patients, its correlation with quality of life and life satisfaction, and its relationship to clinical outcomes in hemodialyzed patients. *Materials and Methods*: Seventy patients with chronic kidney disease undergoing hemodialysis and aged 18 years or older were recruited from a single Dialysis Unit. The Impact of Visual Impairment Scale (IVIS), WHOQOL-BREF, and Cantril Ladder questionnaires were utilized to assess both sociodemographic and clinical variables. *Results*: It was found that, among all assessed variables (i.e., sex, marital status, level of education, months on hemodialysis, history of kidney transplantation, Kt/V, URR, and UF), only age and central venous catheter placement were positively correlated with IVIS scores, while arteriovenous fistula and willingness to become a kidney transplant recipient were negatively correlated. Furthermore, a comparison between patients with moderate and severe visual impairment yielded supplemental data indicating that individuals whose dialysis access was through a dialysis catheter and those ineligible or unwilling to undergo transplantation suffered more often from severe visual impairment. This finding may be attributed to age. *Conclusions*: Older patients were predominantly observed to experience visual impairment. Patients intending to receive a kidney transplant and whose dialysis access was through an arteriovenous fistula were less prone to visual impairment, compared to those who may be ineligible or unwilling to receive transplantation and those with hemodialysis catheters. This phenomenon can be attributed to age-related distinctions in patients’ suitability for specific dialysis access and transplantation. Those reporting visual impairment gave lower ratings in all four domains of their quality of life (comprising physical health, psychological health, social relationships, and environment) and in both present and anticipated five-year life satisfaction. More severe visual impairment was related to an additional reduction in physical health, social relationship, and environment domains, and in life satisfaction.

## 1. Introduction

Chronic kidney disease (CKD) has been observed to have a global prevalence ranging from 3.9% to 15.3% [1]. Patients in stage 5 of CKD are eligible to receive renal replacement therapy, such as renal transplantation, peritoneal dialysis, or primarily hemodialysis (HD) [1,2]. HD therapy necessitates connecting to a machine during sessions and may have a detrimental effect on a patient’s daily life, including prolonged time away from work and family, impairment of family and social life, water and dietary restrictions, and dependence on continuous drug therapy [2,3,4]. The quality of life and life satisfaction of HD patients are also likely to be negatively impacted by HD sessions [2,5]. Moreover, Chiu et al. noted that there have been considerable changes in the demographic characteristics of HD patients, such as increased disability, increased complexity of comorbidities, and an older age at the commencement of HD [6].

The prevalence of visual impairment among older adults is substantial, causing deleterious effects on daily life, social relationships, communication, mobility, and participation in the activities of daily living [5,7,8,9]. This condition is associated with increased mortality rates, higher risk for cardiovascular diseases, and depression in the general population [8]. With the continuous rise in the world’s population, the number of people with visual impairment is expected to augment significantly [10]. Specifically, 28% of individuals aged 70 and older have irreparable visual impairment or blindness [6]. It has been shown that HD patients are at heightened risk of developing visual impairment due to the burden of hypertension and diabetes [6,8], which are both independent risk factors for retinopathy and vision loss [6]. In addition, the risk factors for visual impairment, such as low education status, advanced age, and lack of private health insurance, are similar in the general population and HD patients [9]. Moreover, visual impairment is an indicator of aging and an independent predictor of cardiovascular disease and all-cause mortality in the general population and is an independent risk factor for negative clinical outcomes in HD patients [9]. This shows how important quality of vision is for HD patients and how important the deterioration of vision is as a risk factor.

Presently, the efficacy of treatment is evaluated not only based on the remission of the disease, but also on the improvement of quality of life and life satisfaction [11,12,13,14,15]. HD patients also pay special attention to the quality of their health and lives [16]. It is emphasized that apart from achieving mineral balance, and medical parameters that prove the correctness of HD, such as Kt/V [9], it is also important for healthcare professionals to pay attention to the needs of patients and their quality of life. Healthcare professionals should remember that patients themselves do not care so much about obtaining the correct parameters of HD, but about their quality of life, life satisfaction, and the ability to participate in everyday activities [16]. Therefore, this study was designed and conducted to assess visual impairment in HD patients and its correlation with quality of life and life satisfaction, as well as the associations between visual impairment and clinical outcomes in HD patients.

## 2. Materials and Methods

Our studies assessing quality of life and the various aspects affecting the disease in a group of HD patients were approved by the Bioethical Committee (K.B.Cz. —0014/2017, 18 October 2017) and were conducted between 2017 and 2020 in the Dialysis Unit and the General Surgery Department of the Regional Specialist Hospital. The results provided herein are part of a broader project aimed at evaluating many aspects of HD patients’ lives, with particular emphasis on their own perspectives. The whole project includes, in addition to the visual impairment, quality of life, and life satisfaction indormation presented in this paper, fatigue, sexual dysfunction, cognitive disorders, anxiety, depression, pain, gastrological disorders, and the need for support from the environment, which are very important for this group of patients. This paper presents a single-center cross-sectional study. Patients with chronic kidney disease who were undergoing hemodialysis treatment and aged 18 years or older were recruited from a single Dialysis Unit. The objectives of the study were explained to all participants, and those who provided informed consent were asked to answer the Impact of Visual Impairment Scale (IVIS), World Health Organization Quality of Life-BREF (WHOQOL-BREF), and Cantril Ladder questionnaires. Patients who did not answer the questionnaires, were under the age of 18, refused to participate, or had incomplete data were excluded. At the time of study enrolment, baseline clinical and demographic data were collected from the patients’ medical histories and clinical charts. This included sociodemographic variables (age, sex, marital status, and education) and clinical assessment (months on HD, modality of vascular access, past and desired kidney grafts, fulfillment of medical recommendations, urea reduction ratio (URR), ultrafiltration (UF), and HD adequacy, defined as Kt/V). Of the 136 initially considered HD patients, 70 provided full data on the studied variables and were further evaluated. The demographic characteristics are presented in Table 1, while the clinical characteristics are presented in Table 2.

An evaluation of visual impairment was conducted using the IVIS questionnaire. The IVIS is a five-item self-report questionnaire based on items from the Functional Capacities Assessment developed by the Michigan Commission for the Blind, which assesses difficulties with everyday activities such as watching television, reading, recognizing house numbers, etc. [17]. The total IVIS score ranges from 0 to 15 points, with a higher IVIS score indicating more severe visual impairment [17]. It should be noted, however, that the IVIS does not involve any objective visual examination, nor does it assess the cognitive aspects of vision, such as visual information processing. Additionally, in this study, patients were not subjected to ophthalmological examinations. Patients themselves were asked to rate whether they experienced visual impairments, and whether visual impairments hindered their daily activities. Two questionnaires were selected to analyze the correlation between visual impairment and both quality of life and life satisfaction: the WHOQOL-BREF and Cantril Ladder, respectively. The WHOQOL-BREF, developed by the World Health Organization (WHO), is a gold standard tool for QoL assessment [18,19]. This self-report inventory contains 26 items divided into four domains: physical health, psychological health, social relationships, and environment. Additionally, two separate questions measure overall self-perception of quality of life and overall self-perception of health [18,19,20,21,22]. The items of the WHOQOL-BREF are rated on a 5-point Likert scale, and the domain scores are calculated on a scale of 0−20, where higher scores represent better quality of life. The physical health domain evaluates the ability of patients to perform the activities of daily living, dependence on medicinal aids and substances, pain and discomfort, fatigue and energy levels, mobility, rest, and sleep. The psychological domain assesses bodily image and appearance, negative and positive feelings, self-esteem, spiritual and religious beliefs, cognition, learning, memory, and concentration. The social relationships domain analyzes personal relationships, social support systems, and sexual activity. Finally, the environment domain looks into financial resources, freedom, physical security and safety, access to and quality of healthcare and social care, home environment, opportunities for personal growth and development, the physical environment, and opportunities for recreation and leisure [11,18,19].

General life satisfaction is defined as a subjective, non-disease-specific measure of overall well-being, mental health, and happiness. The Cantril Ladder, a one-item self-report instrument, is used to assess self-reported LS. This simple visual measure presents participants with two ladders, each with 11 rungs numbered from 0 to 10. The top rung represents the best possible life satisfaction, while the bottom rung represents the worst possible life satisfaction. Participants are asked to indicate their current LS on the first ladder, and their expected life satisfaction in five years on the second ladder. The Cantril Ladder assessment is represented by an 11-level Likert score from 0 (lowest life satisfaction) to 10 (highest life satisfaction).

The WHOQOL-BREF and IVIS questionnaires were formally administered under the agreement of the WHO and the National MS Society.

Continuous variables were expressed as the mean and standard deviation (SD), and categorical variables were presented as numbers and percentages. A Student’s *t*-test was used to compare the variables, and an analysis of variance (ANOVA) was performed to compare multiple samples. Statistical significance was determined using a *p*-value of less than 0.05.

## 3. Results

Seventy patients undergoing HD with a mean age of 62.8 years were evaluated, with 46 males (65.7%) included in the sample (Table 1). A *t*-test analysis revealed no significant differences in sex, marital status, level of education, months on HD, length of a single HD session, history of kidney transplant, fulfillment of medical recommendations, Kt/V, URR, UF, and urea concentration between patients who denied visual impairment (IVIS 0 pts) and those who reported impairments (IVIS 1–15 pts) (Table 1 and Table 2). Age and central venous catheter placement were positively correlated, while arteriovenous fistula and willingness to receive a kidney transplant were negatively correlated with IVIS scores (Table 1 and Table 2). When those with moderate visual impairment (IVIS 1–4 pts) were compared to those with severe impairment (IVIS 5–15 pts), it was observed that fewer months on HD, willingness to receive a kidney transplant, lower Kt/V, and URR were associated with severe visual impairment (Table 2).

Finally, the results presented above were supplemented with those obtained from the WHOQOL-BREF and Cantril Ladder questionnaires (Table 3). It was found that patients reporting visual impairment had a lower rate in each domain of the WHOQOL-BREF questionnaire (physical health, psychological health, social relationships, and environment), self-reported quality of life, as well as present and expected life satisfaction in the Cantril Ladder survey. Additionally, separate evaluations of patients reporting severe visual impairment (IVIS 5–15 pts) revealed that they rated lower in the physical health, social relationships, and environment domains of the WHOQOL-BREF questionnaire, as well as present and expected life satisfaction in the Cantril Ladder questionnaire (Table 3).

## 4. Discussion

Our results indicate that age was the only sociodemographic factor significantly correlated with visual impairment. Previous research has described a trend of increasing severity of visual impairment with age in the general population, and visual impairment is known to be an indicator of age and chronic illnesses [7,9]. Nusinovici et al. proposed that visual impairment in HD patients may be due to HD itself or several comorbidities, such as diabetes, hypertension, or anemia, which demonstrate increased frequency in elderly individuals. Moreover, the higher rate of visual impairment found among elderly HD patients could be further explained by the higher frequency of glaucoma and cataracts observed in both general and HD populations with advancing age [8]. In order to more accurately determine the correlation between comorbidities and visual impairment in HD patients, further studies involving larger cohorts are necessary. We did not find any correlations between visual impairment and other sociodemographic factors, indicating that despite higher levels of education and better economic status potentially allowing for access to better medical care, visual impairment can occur regardless of level of education, sex, and marital status [2].

Vascular access in patients undergoing HD may be through either an arteriovenous fistula or a central venous catheter. Catheters are more commonly used in patients of advanced age and those with a short life expectancy [2]. This is reflected in the higher percentage of patients with catheters among those with visual impairment, as reported in Table 2. Visual impairment is a potential indicator of age and chronic illness, and is thought to be a direct factor in increased mortality [9]. Given that elderly patients are commonly treated with HD through a dialysis catheter, the presence of a correlation between catheter-initiated dialysis and visual impairment should not be unexpected. Hong et al. already reported the abovementioned correlation between visual impairment and comorbidities such as diabetes mellitus and cardiovascular diseases [9]. In addition, HD patients with visual impairment had a significantly greater risk of cardiovascular hospitalization compared to those without visual impairment, and cardiovascular diseases are the leading cause of mortality in HD patients [9]. Nephrologists should be cognizant of possible visual impairment in HD patients with dialysis catheters, as they are often elderly or multi-morbid. Therefore, this population of visually impaired patients should receive additional healthcare support, in order to ensure comprehensive and optimal care.

When the time on HD was evaluated, no difference was found between the percentage of patients who reported and did not report visual impairment (Table 2). However, when patients with the most severe visual impairment (IVIS scores of 5–15 points) were evaluated separately, a correlation was observed between the severity of visual impairment and commencing HD treatment within the last 30 months. This may indicate that those with the most severe visual impairment receive HD for a relatively short period due to age, general condition, and comorbidities, which are related to visual impairment [5,7,9].

We have ruled out a correlation between IVIS score and kidney transplant in the past. However, our analysis revealed a significant correlation between willingness to receive a kidney transplant in the future and IVIS score (Table 2). The percentage of patients who were either on the waiting list or wanted to be on the waiting list for a transplant was lower within the visually impaired population, particularly in cases of severe visual impairment (IVIS scores of 5–15 points). It is important to note that those who were not qualified for a transplant and were not on a waiting list were typically elderly adults with concomitant chronic medical conditions that excluded them from transplantation. Elsman et al. described that severe visual impairment can result in social exclusion [7]. Patients undergoing expensive and sophisticated treatments may experience psychological and physical distress, with the uncertainty of their future and physical and financial limitations leading to various depressive symptoms, and a decline in quality of life and life satisfaction [2,3,23,24,25]. These findings may explain why HD patients who rate their quality of life and life satisfaction significantly lower, and expect it to be even worse in the next five years, do not want to opt for a kidney transplant, which would be the best treatment for kidney failure [20,22]. By accurately identifying HD patients who may be ineligible or unwilling to receive kidney transplantation, healthcare professionals can provide more timely diagnoses and treatment, as well as improved support. An ophthalmological examination should be conducted to evaluate visual acuity, in addition to an IVIS test to determine the effect of vision impairment on daily living activities.

Chronically ill patients, including those undergoing HD, were found to have significantly decreased quality of life [3,23,25,26]. These patients experience deterioration in their physical, social, and environmental well-being, as well as their overall mental health. They report increased dissatisfaction with different aspects of their environment, finances, opportunities for recreation, and everyday living [3]. Our results show that HD patients reporting visual impairment experience deterioration in each of the four domains making up the WHOQOL-BREF questionnaire, as well as the single-item question on self-perception of quality of life (Table 3). Consequently, our findings are in line with those of other studies investigating the quality of life of chronically ill individuals. Individuals with severe visual impairment (IVIS 5–15 pts) score even worse in the physical health, social relationships, and environment domains. Muller et al. described that social isolation in chronically ill patients decreases environmental quality of life [11]. This phenomenon was also confirmed in the current study, which points out the correlation between visual impairment and the WHOQOL-BREF environmental domain. Visual impairment has been described as leading to social isolation and difficulty in initiating relationships [7]. Individuals who have fewer friends are more likely to feel lonely and experience a decreased perception of quality of life in this field [2,7,11]. Patients undergoing HD who attend a three-times-weekly dialysis station for several hours have been observed to suffer from limited opportunities to establish and maintain social relationships. In our study, patients with varying degrees of visual impairment indicated commensurate deterioration in the social relationships domain of the self-reported quality-of-life assessments. These findings demonstrate the substantial effect of visual impairment on a patient’s quality of life, as well as their capacity to engage with their environment.

Significantly lower quality of life has previously been reported in CKD patients compared to matched controls [26]. This study observed deterioration in the quality of life of HD patients that correlated with the presence of visual impairment. In analyses comparing those who reported visual impairment to those who did not, it was found that those with visual impairment rated their quality of life as worse in the physical, psychological, social relationships, and environmental domains, with the greatest decrease in the environmental domain. However, when evaluating self-perceived sense of health, there were no significant differences between those with and without visual impairments (Table 3). This observation supports the assertion found in the literature [3] that chronic diseases, such as end-stage renal failure requiring HD, have a detrimental effect on the quality of life of patients. No significant impact on perceived health, however, was observed. An analysis of the single-item Cantril Ladder questionnaire was conducted to investigate the correlation between visual impairment and life satisfaction [22,27]. Life satisfaction is a product of a variety of factors, such as prior experiences, existing circumstances, and expectations regarding the future. Positive emotion, positive self-image, and socioeconomic factors are major predictors of Cantril Ladder scale scores [4]. Our findings indicated that both current and anticipated life satisfaction in five years were worse for visually impaired HD patients (IVIS 1–15 pts), particularly those who reported severe impairment (IVIS 5–15 pts). This is the same cohort of HD patients who self-reported poorer quality of life. In contrast, our research revealed that the self-assessment of general health was independent of visual impairment. Mazur et al. documented only a weak connection between life satisfaction and physical health in a healthy adolescent population [4]. This phenomenon could account for why HD patients who reported visual impairment also reported lower life satisfaction and quality of life, but not general health. The literature reports that HD patients attach a great amount of value to quality of life and the ability to reach personal goals [16]. These parameters are highly important to patients, who often use them to judge the success of their overall treatment. Quality of life and life satisfaction are viewed as being more crucial to patients than laboratory outcomes and measures of HD efficiency [16].

Nusinovici et al. suggested that the high prevalence of visual impairment among CKD patients is likely due to the similarity of a network of vessels in the glomerulus and choroid, as well as between the glomerular filtration barrier and inner retina [8]. The mechanisms underlying both CKD and ocular diseases, such as atherosclerosis, inflammation, endothelial dysfunction, and oxidative stress, are similar [8,9]. Thus, we can support the suggestion of Nusinovici et al. [8] that it should be recommended that renal transplant recipients undergo a pretransplantation examination by an ophthalmologist. Additionally, we hypothesize that regular eye examinations, both before and during HD therapy, should be encouraged for all CKD patients to reduce the negative effects of visual impairments and improve the early detection and treatment of eye diseases [9]. In order to ensure the appropriate management of HD patients, interdisciplinary cooperation between ophthalmologists and nephrologists is essential [8]. Consequently, the responsibility of healthcare administrators should be to guarantee that patients receive care in a way that enables them to be routinely assessed, and that treatment can be initiated at the most advantageous time.

Visual impairment has been associated with a range of psychological changes, including cognitive impairment, isolation, and depression [9]. There is a strong correlation between any degree of visual sensation impairment and decreased quality of life [6]. The identification of and tailored attention to the needs of those individuals with HD and visual impairment may mitigate the effect of the disease on their quality of life [9,28].

Our study has several limitations, including its single-center design and exclusion of participants under 18 years of age. Furthermore, visual acuity was self-reported via questionnaire and not evaluated by an ophthalmologist. We did not account for the type of eye disease, such as glaucoma, cataract, diabetic retinopathy, or age-related macular degeneration. Additionally, the number of patients who withdrew from the study was relatively high, with 66 patients excluded out of the initial 136 considered. This research suggests potential avenues for further exploration, such as the evaluation of subjective visual deterioration in patients based on ophthalmological examination results and the examination of juvenile patients.

## 5. Conclusions

Older patients are predominantly observed to experience visual impairment. Patients intending to receive a kidney transplant and whose dialysis access is through an arteriovenous fistula are less prone to visual impairment, compared to those who may be ineligible or unwilling to receive transplantation and those with HD catheters. This phenomenon can be attributed to age-related distinctions in patients’ suitability for specific dialysis access and transplantation. Those reporting visual impairment give lower ratings in all four domains of their quality of life (comprising physical health, psychological health, social relationships, and environment) and both present and anticipated five-year life satisfaction. More severe visual impairment is related to an additional reduction in the physical health, social relationship, and environment domains, and life satisfaction. The results of the study suggest that healthcare professionals should take into consideration ophthalmological examinations for HD patients, particularly those who are elderly. Administration of the IVIS questionnaire to patients enables the identification of those individuals whose visual impairment may adversely impact their daily activities and overall quality of life. It is prudent to consider an ophthalmological examination of each patient initiating HD treatment.

## Figures and Tables

**Table 1 medicina-59-01106-t001:** The sociodemographic characteristics of hemodialyzed patients and the Impact of Visual Impairment Scale (IVIS) scoring.

			IVIS Score											
			0 pts			1 pt−15 pts			1 pt−4 pts			5 pts−15 pts		
		*n* (%) M [Range] (SD)	*n* (%) M [Range] (SD)	Vs. Population *p*-Value	Vs. Sample *p*-Value	*n* (%) M [Range] (SD)	Vs. Population *p*-Value	Vs. Sample *p*-Value	*n* (%) M [Range] (SD)	Vs. Population *p*-Value	Vs. Sample *p*-Value	*n* (%) M [Range] (SD)	Vs. Population *p*-Value	Vs. Sample *p*-Value
		70 (100)	32 (45.7)			38 (54.3)			18 (25.7)			20 (28.6)		
Sex	Male	46 (65.7)	21 (65.6)	N/S	N/S	25 (65.8)	N/S	N/S	12 (66.7)	N/S	N/S	13 (65.0)	N/S	N/S
	Female	24 (34.3)	11 (34.4)	N/S	N/S	13 (34.2)	N/S	N/S	6 (33.3)	N/S	N/S	7 (35.0)	N/S	N/S
Age (yrs)		62.8 [28–84] (12.4)	59.1 [28–78] (12.1)	↓ <0.02	↓ <0.02 vs. “1 pt–15 pts” ↓ <0.002 vs. “5 pts–15 pts”	66.0 [40–84] (12.0)	↑ <0.03	↑ <0.02 vs. “0 pts”	61.1 [40–81] (13.0)	N/S	↓ <0.02 vs. “1 pt–4 pts”	7.4 [54–84] (9.2)	↑ <0.0004	↑ <0.002 vs. “0 pts” ↑ <0.02 vs. “1 pt–4 pts”
Marital status	Married	48 (69.6)	20 (64.5)	N/S	N/S	28 (73.7)	N/S	N/S	14 (77.8)	N/S	N/S	14 (70.0)	N/S	N/S
	Not-married	21 (30.4)	11 (35.5)	N/S	N/S	10 (26.3)	N/S	N/S	4 (22.2)	N/S	N/S	6 (30.0)	N/S	N/S
Education level	Primary school	38 (55.1)	14 (45.2)	N/S	N/S	24 (63.2)	N/S	N/S	10 (55.6)	N/S	N/S	14 (70.0)	N/S	N/S
	Secondary school	19 (27.5)	11 (35.5)	N/S	N/S	8 (21.0)	N/S	N/S	4 (22.2)	N/S	N/S	4 (20.0)	N/S	N/S
	Higher education	12 (17.4)	6 (19.3)	N/S	N/S	6 (15.8)	N/S	N/S	4 (22.2)	N/S	N/S	2 (10.0)	N/S	N/S

N/S—*p* > 0.05.

**Table 2 medicina-59-01106-t002:** The clinical characteristics of hemodialysis patients and the Impact of Visual Impairment Scale (IVIS) scores.

				IVIS Score											
				0 pts			1 pt−15 pts			1 pt−4 pts			5 pts−15 pts		
			*n* (%) M [Range] (SD)	*n* (%) M [Range] (SD)	Vs. Population *p*-Value	Vs. Sample *p*-Value	*n* (%) M [Range] (SD)	Vs. Population *p*-Value	Vs. Sample *p*-Value	*n* (%) M [Range] (SD)	Vs. Population *p*-Value	Vs. Sample *p*-Value	*n* (%) M [Range] (SD)	Vs. Population *p*-Value	Vs. Sample *p*-Value
Vascular access	AVF		60 (87.0)	30 (96.8)	N/S	↑ <0.05 vs. “1 pt–15 pts” ↑ <0.04 vs. “1 pt–4 pts”	30 (81.1)	N/S	↓ <0.05 vs. “0 pts”	14 (77.8)	N/S	↓ <0.04 vs. “0 pts”	16 (84.2)	N/S	N/S
	CVC		8 (13.0)	1 (3.2)	N/S	↓ <0.05 vs. “1 pt–15 pts” ↓ <0.04 vs. “1 pt–4 pts”	7 (18.9)	N/S	↑ <0.05 vs. “0 pts”	4 (22.2)	N/S	↑ <0.04 vs. “0 pts”	3 (15.8)	N/S	N/S
Months on HD			46.2 [1–317] (61.1)	53.0 [1–317] (78.6)	N/S	N/S	4.6 [1–204] (41.8)	N/S	N/S	51.0 [2–204] (48.8)	N/S	N/S	3.1 [1–108] (31.3)	↓ <0.02	N/S
Length of HD session (hrs)			3.94 (0.34)	3.93 (0.33)	N/S	N/S	3.94 (0.36)	N/S	N/S	4.02 (0.40)	N/S	N/S	3.87 (0.32)	N/S	N/S
Kidney transplant in the past		Yes	5 (7.0)	3 (9.4)	N/S	N/S	2 (5.3)	N/S	N/S	2 (11.1)	N/S	N/S	0 (0)	N/S	N/S
		No	65 (93.0)	29 (90.6)	N/S	N/S	36 (94.7)	N/S	N/S	16 (88.9)	N/S	N/S	20 (100.0)	N/S	N/S
Desire to receive kidney transplant		Yes	35 (52.2)	21 (67.7)	N/S	↑ <0.03 vs. “1 pt–15 pts” ↑ <0.02 vs. “5 pts–15 pts”	14 (38.9)	N/S	↓ <0.03 vs. “0 pts”	8 (47.1)	N/S	N/S	6 (31.6)	N/S	↓ <0.02 vs. “0 pts”
		No	32 (47.8)	10 (32.3)	N/S	↓ <0.03 vs. “1 pt–15 pts” ↓ >0.02 vs. “5 pts–15 pts”	22 (61.1)	N/S	↑ <0.03 vs. “0 pts”	9 (52.9)	N/S	N/S	13 (68.4)	N/S	↑ <0.02 vs. “0 pts”
Fulfillment of medical recommendations		Yes	44 (67.7)	22 (71.0)	N/S	N/S	22 (66.7)	N/S	N/S	10 (62.5)	N/S	N/S	12 (70.6)	N/S	N/S
		No	20 (33.3)	9 (29.0)	N/S	N/S	11 (33.3)	N/S	N/S	6 (37.5)	N/S	N/S	5 (29.4)	N/S	N/S
Kt/V			1.30 [0.82–3.11] (0.33)	1.31 [0.84–1.88] (0.22)	N/S	N/S	1.29 [0.82–3.11] (0.40)	N/S	N/S	1.43 [0.90–3.11] (0.51)	N/S	↑ <0.04 vs. “5 pts–15 pts”	1.18 [0.82–1.63] (0.22)	↓ <0.01	↓ <0.04 vs. “1 pt–4 pts”
URR			0.660 [0.51–0.90] (0.074)	0.669 [0.53–0.80] (0.062)	N/S	↑ <0.05 vs. “5 pts–15 pts”	0.652 [0.51–0.90] (0.084)	N/S	N/S	0.682 [0.55–0.90] (0.092)	N/S	↑ <0.03 vs. “5 pts–15 pts”	0.625 [0.51–0.74] (0.067)	↓ <0.02	↓ <0.05 vs. “0 pts” ↓ <0.03 vs. “1 pt–4 pts”
UF (mL)			2293 [300–4200] (911)	2216 [500–3500] (905)	N/S	N/S	2361 [300–4200] (924)	N/S	N/S	2253 [300–4200] (1206)	N/S	N/S	2458 [1500–3600] (589)	N/S	N/S
Urea concentration before HD			127.3 (30.9)	121.7 (28.3)	N/S	N/S	132.3 (32.7)	N/S	N/S	134.1 (28.6)	N/S	N/S	13.8 (36.7)	N/S	N/S
Urea concentration after HD			43.1 (13.8)	4.4 (12.7)	N/S	N/S	45.4 (14.6)	N/S	N/S	42.3 (15.2)	N/S	N/S	48.3 (13.8)	N/S	N/S

N/S—*p* > 0.05.

**Table 3 medicina-59-01106-t003:** The correlation between the scores obtained from the Impact of Visual Impairment Scale (IVIS), WHOQOL-BREF, and Cantril Ladder questionnaires.

			IVIS Score											
			0 pts			1 pt–15 pts			1 pt–4 pts			5 pts–15 pts		
		*n* (%) M [Range] (SD)	*n* (%) M [Range] (SD)	Vs. Population *p*-Value	Vs. Sample *p*-Value	*n* (%) M [Range] (SD)	Vs. Population *p*-Value	Vs. Sample *p*-Value	*n* (%) M [Range] (SD)	Vs. Population *p*-Value	Vs. Sample *p*-Value	*n* (%) M [Range] (SD)	Vs. Population *p*-Value	Vs. Sample *p*-Value
IVIS		3.17 [0–15] (4.24)	0 (0)			5.84 [1–15] (4.18)			2.50 [1–4] (1.15)			8.85 [5–15] (3.56)		
WHOQOL-BREF	Physical health	11.84 (1.75)	12.46 (1.56)	↑ <0.005	↑ <0.006 vs. “1 pt–15 pts” ↑ <0.006 vs. “5 pts–15 pts”	11.31 (1.74)	↓ <0.009	↓ <0.006 vs. “0 pts”	11.59 (1.97)	N/S	N/S	11.06 (1.51)	↓ <0.02	↓ <0.006 vs. “0 pts”
	Psychological health	12.55 (2.18)	13.17 (2.21)	↑ <0.05	↑ <0.03 vs. “1 pt–15 pts”	12.03 (2.04)	↓ <0.03	↓ <0.03 vs. “0 pts”	12.10 (1.86)	N/S	N/S	11.97 (2.23)	N/S	N/S
	Social relationships	13.28 (3.20)	14.46 (2.75)	↑ <0.003	↑ <0.005 vs. “1 pt–15 pts” ↑ 0.0398 vs. “1 pt–4 pts” ↑ <0.01 vs. “5 pts–15 pts”	12.28 (3.24)	↓ <0.009	↓ <0.005 vs. “0 pts”	12.52 (2.94)	N/S	↓ <0.04 vs. “0 pts”	12.07 (3.55)	↓ <0.0893	↓ <0.01 vs. “0 pts”
	Environment	13.3 (2.37)	14.45 (2.03)	↑ <0.0002	↑ <0.0001 vs. “1 pt–15 pts” ↑ <0.0001 vs. “5 pts–15 pts”	12.38 (2.25)	↓ <0.0005	↓ <0.0001 vs. “0 pts”	13.28 (1.79)	N/S	↑ <0.02 vs. “5 pts–15 pts”	11.58 (2.35)	↓ <0.0009	↓ <0.0001 vs. “0 pts” ↓ <0.02 vs. “1 pt–4 pts”
	Self-perception of QOL	3.3 [2–4] (0.7)	3.6 [2–4] (0.6)	↑ <0.0006	↑ <0.02 vs. “1 pt–15 pts” ↑ <0.02 vs. “1 pt–4 pts” ↑ <0.03 vs. “5 pts–15 pts”	3.2 [2–4] (0.7)	N/S	↓ <0.02 vs. “0 pts”	3.1 [2–4] (0.8)		↓ <0.02 vs. “0 pts”	3.2 [2–4] (0.6)		↓ <0.03 vs. “0 pts”
	Self-perception of health	2.3 [1–5] (1.0)	2.3 [1–4] (0.9)	N/S	N/S	2.3 [1–5] (1.1)	N/S	N/S	2.4 [1–5] (1.1)	N/S	N/S	2.1 [1–5] (1.1)	N/S	N/S
Cantril Ladder	CL 0	6.10 (2.15)	6.82 (1.98)	↑ <0.01	↑ <0.008 vs. “1 pt–15 pts” ↑ <0.0008 vs. “5 pts–15 pts”	5.47 (2.13)	↓ <0.02	↓ <0.008 vs. “0 pts”	6.12 (1.93)	N/S	↑ <0.05 vs. “5 pts–15 pts”	4.73 (2.15)	↓ <0.004	↓ <0.0008 vs. “0 pts” ↓ <0.05 vs. “1 pt–4 pts”
	CL 5	4.75 (2.71)	5.96 (2.54)	↑ <0.001	↑ <0.0005 vs. “1 pt–15 pts” ↑ <0.05 vs. “1 pt–4 pts” ↑ <0.0001 vs. “5 pts–15 pts”	3.79 (2.47)	↓ <0.002	↓ <0.0005 vs. “0 pts”	4.47 (2.72)	N/S	↓ <0.05 vs. “0 pts”	3.06 (2.02)	↓ <0.0003	↓ <0.0001 vs. “0 pts”

IVIS—Impact of Visual Impairment Scale; WHOQOL-BREF—World Health Organization Quality of Life questionnaire; CL—Cantril Ladder questionnaire (CL 0—life satisfaction at a particular time, CL 5— expected life satisfaction in 5 years). N/S—*p* > 0.05.

## Data Availability

The data presented in this study are available on request from the corresponding author. The data are not publicly available due to privacy restrictions.

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
