# Peer review of "Visual Impairment in Hemodialyzed Patients—An IVIS Study"

_medicina, 2023, doi:10.3390/medicina59061106_

Round 1
Reviewer 1 Report
I would like to begin by greeting the authors and congratulating them for having decided to investigate an area where there is still so much to discover, but also for having decided to share this article with the rest of the scientific community, so that science can evolve.
This is a study on visual impairment in hemodialyzed patients. It is a very important topic, current in permanent evolution and which it never hurts to have more information.
All comments, doubts and suggestions made are constructive and try to improve the article, after several attentive readings.
Title
I believe that the title is presented too simply, not giving an indication of what was evaluated or the type of study that was used.
Abstract
Authors must follow the rules defined by the journal for writing the Abstract:
“This is the abstract section, about 300 words maximum. For articles research, systematic reviews or meta-analyses, abstracts should give a relevant overview of the work. We strongly encourage authors to use the given subheadings. Background and Objectives: Place the question addressed in a broad context and highlight the purpose of the study. Materials and Methods: Briefly describe the main methods or treatments applied, including the study population description. Results: Summarize the article's main findings. Conclusions: Indicate the main conclusions or interpretations. The abstract should be an objective representation of the article, it must not contain results which are not presented and substantiated in the main text and should not exaggerate the main conclusions.”
The results section of the abstract should present concrete results of the study and not just generalities.
In the conclusions section of the abstract I would like to see some indications about clinical practice. How should this article change clinical practices and what other studies can or should be done?
Keywords:
Repetitions with expressions that are in the title should be avoided. Whenever possible keywords should be Mesh. Authors should review the keywords.
Introduction
The objective of the end of the Introduction must be exactly the same as that of the Abstract and vice versa.
In the introduction, the authors must make a clear reference to the reason why this article should be published in a journal as important as Medicina. What data from this article can change the practice of nephrologists around the world? What does this article bring to scientific knowledge? Are there clinical practices that should be changed? What kind of studies are still needed?
Materials and methods
What is the type of study? Authors must clearly indicate in the text.
Data were collected between 2017 and 2020. The reason for this time gap must be explained.
Has the broader project referred to in the materials and methods section been published? If so, it must be referenced.
Results
Tables are not formatted according to the Medicina Template
Conclusions
“Patients intending to receive a kidney transplant and whose dialysis accesses are arteriovenous fistulas are less prone to visual impairment”. Is this a cause or a consequence? In other words, are these patients less likely to have reduced visual acuity, or is it because they are less likely to have reduced visual acuity that they have arteriovenous fistulas and are on the transplant waiting list?
The results of this study are important because they allow, in terms of clinical practice, what? What can change in terms of the approach of nephrologists around the world?
General Comments
Well written article that I enjoyed reading.
Quality of English adequate. Minor editing of English language required
Author Response
Dear Reviewer,
Thank you for taking the time to review our paper. We appreciate the insight and guidance given, which enabled us to further enhance its substantial value.
In response to the comments provided, our article's Title, Abstract, and Keywords have been edited and modified accordingly. Additionally, the Introduction was expanded to include a clarification of the clinical implications of the research and its potential utilization in practice. In the Materials and Methods section, we clarified the type of study conducted and elaborated on its duration. The Tables were also formatted according to the precise requirements of the Medicina Template, and are attached as separate files. Lastly, we have included substantial excerpts within the Discussion and Conclusions sections to explain the importance of our research in the context of medical practice and how it could potentially improve the quality of life of dialysis patients. Moreover, the entire document has been thoroughly checked for English accuracy.
We hope that the revised version of the article meets the standards for publication in the Medicina journal.
Respectfully,
Authors
Reviewer 2 Report
Thank you for reviewing this interesting manuscript. The author hypothesizes that patients intending to receive a kidney transplant and whose dialysis accesses are arteriovenous fistulas are less prone to visual impairment. Their finding suggests that visual impairment does not develop with an arteriovenous fistula, unlike a dialysis catheter, even if no kidney transplantation occurs. However, this is not true, so I do not recommend publishing the manuscript in its current form. It is self-evident that visual impairment causes a further decrease in quality of life and outcome; this is nothing new.
Furthermore, the material and methods section also needs improvement; there is no place for references here.
Author Response
Dear Reviewer,
We are thankful for taking the time to review our paper, as it has enabled us to analyze our research from a different standpoint and augment its substantive value.
In accordance with the reservations expressed in the review, we have amended the article. Specifically, the Introduction has been supplemented with a clarification on the clinical implications of our research and how they could be applied clinically. The Discussion and Conclusions sections were further elucidated with explanations as to why our research is relevant for clinical practice and how it could alter the care of dialysis patients to improve their quality of life. We have also edited the Materials and Methods section, reducing the number of references, leaving only those referring to diagnostics questionnaires utilized. Additionally, the work has been exhaustively examined for English language accuracy.
We hope that the indicated revisions have conformed to the standards set by the Medicina journal.
Respectfully,
Aurthors
Reviewer 3 Report
Dear authors,
Your article proposal touch an interesting and actual subject. Patients with chronic kidney disease (CKD) receiving dialysis have higher mortality and morbidity compared to the general population. Visual impairment has long been recognized as an important factor in the aging process, and there is increasing awareness of its influence on health and functional status, particularly in CKD. Visual impairment often limits people's ability to perform daily tasks and affects their quality of life.
Your's primary objective was to demonstrate visual impairment in HD patients and its correlation with quality of life and life satisfaction. The paper is well written, in accordance with the rules of the journal. Material section can be improved. The discussion section is well written, it has the limits of the study. The conclusions are consistent with the results.
References can be improved. This is a good and interesting topic for nephrologist, ophthalmologist, general practitioner, psychologist, so there are many articles on this topic.
Good luck!
Minor editing of English language required
Author Response
Dear Reviewer,
We would like to thank you for taking the time to review our paper and providing us with the opportunity to improve its substantive value.
We addressed the issues you raised in your review. We have made corrections to the Introduction, Materials and Methods, Discussions and Conclusions, and Literature sections. We have added explanations of the clinical inclinations of the research and how they could be used in clinical work, marked the type of study and its duration, and provided excerpts explain how our research is relevant to clinical practice and how it could improve the quality of life of dialysis patients. We have also checked for the correctness of the English language.
We hope that these corrections meet the requirements set by the Medicina journal.
Respectfully,
Authors
Round 2
Reviewer 2 Report
The authors almost answered the raised questions. The manuscript definitely improved.
I have no more questions.